# Wnt/β-catenin signaling regulates amino acid metabolism through the suppression of CEBPA and FOXA1 in liver cancer cells
Saya Nakagawa [1], Kiyoshi Yamaguchi [1]✉, Kiyoko Takane[1], Sho Tabata [2], Tsuneo Ikenoue[1] & Yoichi Furukawa [1]✉

Deregulation of the Wnt/β-catenin pathway is associated with the development of human cancer including colorectal and liver cancer. Although we previously showed that histidine ammonia lyase (*HAL*) was transcriptionally reduced by the β-catenin/TCF complex in liver cancer cells, the mechanism(s) of its down-regulation by the complex remain to be clarified. In this study, we search for the transcription factor(s) regulating *HAL*, and identify CEBPA and FOXA1, two factors whose expression is suppressed by the knockdown of β-catenin or TCF7L2. In addition, RNA-seq analysis coupled with genome-wide mapping of CEBPA- and FOXA1-binding regions reveals that these two factors also increase the expression of arginase 1 (ARG1) that catalyzes the hydrolysis of arginine. Metabolome analysis discloses that activated Wnt signaling augments intracellular concentrations of histidine and arginine, and that the signal also increases the level of lactic acid suggesting the induction of the Warburg effect in liver cancer cells. Further analysis reveals that the levels of metabolites of the urea cycle and genes coding its related enzymes are also modulated by the Wnt signaling. These findings shed light on the altered cellular metabolism in the liver by the Wnt/β-catenin pathway through the suppression of liver-enriched transcription factors including CEBPA and FOXA1.

Alterations in the genes associated with the Wnt/β-catenin signaling pathway have been identified in various types of tumors[1]. In colorectal cancer, over 90% of the cases carry at least one mutation in genes involved in this pathway such as inactivating mutations of the adenomatous polyposis coli (*APC*) gene (~80%) and activating mutations of the β-catenin (*CTNNB1*) gene (~5%)[2]. In hepatocellular carcinoma (HCC), somatic mutations in *CTNNB1* (31%) and *AXIN1* (6%) have been frequently found[3]. These mutations result in the accumulation of β-catenin and its translocation into the nucleus to form transcriptionally active complexes with T-cell factor/lymphoid enhancer factor (TCF/LEF) family proteins. Importantly, genes directly transactivated by the complexes have been shown to play a key role in the development of tumors. For example, c-Myc and cyclin D1 regulate cell proliferation and/or cell cycle progression[4,5]. Although more than one hundred target genes have been identified, most of the genes are transcriptionally up-regulated by the complex, and little attention have been paid to the down-regulated genes.

We previously reported that the suppressed expression of IRF1 by the signal plays a vital role in colorectal carcinogenesis, and its expression was regulated by the ubiquitin-proteasome pathway through a deubiquitinase complex, USP1/UAF1[6]. We also revealed that histidine ammonia-lyase (*HAL*), phosphoenolpyruvate carboxykinase 1 (*PCK1*), solute carrier family 51 subunit alpha (*SLC51A*), pleckstrin 2 (*PLEK2*), integrin β3 (*ITGB3*), and secretory leukocyte protease inhibitor (*SLPI*) were transcriptionally down-regulated by the Wnt signaling pathway in liver cancer[7]. Additionally, we identified a regulatory region of *HAL*, between −90 bp and −44 bp in the 5'-flanking region, and revealed that transcriptional activity of the region was suppressed by the activation of Wnt/β-catenin signaling[7].

HAL is the rate-limiting enzyme of histidine catabolism and catalyzes L-histidine to urocanate and ammonia in the liver and skin. Histidine metabolism is an important metabolic process involved in de novo synthesis of purine nucleic acids that accelerates the cancer cell progression due to the association with production of tetrahydrofolate[8]. In addition, histamine, a metabolite converted from histidine, plays an important role in cancer

[1]Division of Clinical Genome Research, Advanced Clinical Research Center, The Institute of Medical Science, The University of Tokyo, Tokyo 108-8639, Japan. [2]Tsuruoka Metabolomics Laboratory, National Cancer Center, Tsuruoka, Yamagata 997-0052, Japan. ✉e-mail: kiyamagu@g.ecc.u-tokyo.ac.jp; yofurukawa@g.ecc.u-tokyo.ac.jp

https://doi.org/10.1038/s42003-024-06202-9 **Article**

immunity through the control of various responses in innate and adaptive immunity[9]. Although the expression of HAL may play a crucial role in cancer cells, the regulatory mechanism remains to be clarified.

Regarding the link between the canonical Wnt signaling and metabolism, it was reported that the activation of the Wnt signaling pathway induced the Warburg effect through increased expression of $MYC$[10], $PDK1$[11], and lactate/pyruvate transporter $MCT1/SLC11A1$[12] or suppression of cytochrome oxidase[13]. Importantly, crosstalk between the Wnt and c-Myc pathways promotes glycolysis and energy production in cancer cells[14]. In addition, glycolysis modulates Wnt signaling to promote axial elongation of the embryo in the tail bud[15], suggesting a tight link between Wnt signaling and aerobic glycolysis. However, little is known about the effect of Wnt signaling on other metabolisms.

In this study, we clarified that $HAL$ was transcriptionally induced by CEBPA and FOXA1, and that their expression levels were down-regulated by the Wnt/β-catenin signaling in the liver cancer cells. Furthermore, we found that activated Wnt/β-catenin signaling suppressed not only $HAL$ but also $ARG1$, a gene encoding arginase 1, through the reduced CEBPA and FOXA1 expression. These data helped us gain insight into a role of the signal pathway in amino acid metabolism in liver cancer.

## Results

### Identification of candidate transcription factor(s) regulating *HAL* in liver cancer cells

We previously uncovered that the 5'-flanking region of $HAL$, between −90 bp and −44 bp, was responsible for the suppression of β-catenin/TCF complex in liver cancer cells[7]. Since the β-catenin/TCF complex induces downstream target genes, the decreased transcription of $HAL$ was assumed to be indirectly regulated by the complex. In agreement with this notion, the region has no β-catenin/TCF7L2 ChIP-seq peaks or consensus TCF-binding motifs (ENCODE accession number: ENCSR000EVQ).

We searched for transcription factors (TFs) that are associated with the activity of the $HAL$ 5'-flanking region using JASPAR, a database of TF-binding profiles (http://jaspar.genereg.net/). As a result, a total of 41 putative motifs including 33 TFs with a score of 8.0 or above were identified in the region (Supplementary Data 1a). We searched for two types of TFs among the 33 identified, namely transcriptional repressors that were up-regulated by the Wnt/β-catenin pathway and activators that were down-regulated by the pathway. Expression profile analysis was performed using HepG2 hepatoblastoma cells transfected with different siRNAs targeting β-catenin (−9 or −10), TCF7, TCF7L1, TCF7L2, LEF1, or their combination (Supplementary Data 1b, c) and the expression levels of the 33 TFs were compared with those of downstream genes in the Wnt signaling pathway using the expression profile data (Supplementary Fig. 1a). As expected, $MYC$, $RNF43$, $AXIN2$, and $LGR5$, four well-known Wnt target genes were down-regulated by the treatment with β-catenin, TCF7L2, or TCF7 siRNA (Fig. 1a). However, TCF7L1 and LEF1 are unlikely to be involved in the activation of Wnt signaling in the cells, possibly due to the lower expression of $TCF7L1$ and $LEF1$ compared with other TCF family members in HepG2 cells. We additionally confirmed that $HAL$ expression was remarkably enhanced by the treatment with β-catenin, TCF7L2, or TCF7 siRNA.

A hierarchical clustering analysis using expression values of the 33 TFs, $HAL$, and the four Wnt target genes identified subsets of genes that exhibited different expression patterns in response to the suppression of Wnt signaling (Fig. 1a). Intriguingly, $FOXD1$ and $FOXP1$ were classified in a subgroup containing the four Wnt target genes ($MYC$, $RNF43$, $AXIN2$, and $LGR5$), suggesting that these two genes may be up-regulated by the activation of Wnt signaling. FOXD1 is known to be a transcriptional activator[16], but FOXP1 was reported to act primarily as a transcriptional repressor[17,18]. Therefore, we included FOXP1 as a candidate suppressor of $HAL$. Additional qPCR analysis confirmed that $FOXP1$ expression was significantly decreased by the depletion of β-catenin- or TCF7L2 in HepG2 cells (Supplementary Fig. 1b). Although we treated HuH-7 cells with two-independent FOXP1 siRNA, $HAL$ expression was not induced by the

treatment (Supplementary Fig. 1c), suggesting that FOXP1 may not suppress the transcription of $HAL$ in the cells.

We additionally identified nine TFs including $CEBPA$, $FOXA1$, $FOXA3$, $FOXK1$, $FOXN3$, $FOXP3$, $NFATC2$, $PROP1$, and $SRY$, which were classified in a subgroup containing $HAL$ by the cluster analysis, suggesting that these TFs are candidate transcriptional activator(s) regulating $HAL$. Among the nine TFs, we focused on factors whose expression was correlated with $HAL$ in liver cancer tissues using the TCGA data (cBioportal, https://www.cbioportal.org/), because the negative regulation of $HAL$ by Wnt signaling was most explicit in liver cancer, but not in colorectal cancer[7]. As a result, the expression levels of $FOXA3$ ($r = 0.36$), $FOXA1$($r = 0.25$), and $CEBPA$ ($r = 0.22$) were significantly correlated with those of $HAL$ ($q$-value < 0.01, Supplementary Fig. 2a). To validate the association of their expression with the Wnt signaling, we carried out qPCR and western blot analyses using HepG2 and HuH-6 cells carrying activating mutations in the $CTNNB1$ (β-catenin) gene. As expected, silencing of β-catenin or TCF7L2 increased the expression of CEBPA, FOXA1, and FOXA3 at the RNA and protein levels (Fig. 1b, c and Supplementary Fig. 2b). It is of note that these three TFs are known as liver-enriched TFs[19], and that $HAL$ and the three TFs are also abundantly expressed in normal liver tissue (Supplementary Fig. 2c, d). These data suggested that CEBPA, FOXA1, and FOXA3 were candidates that transcriptionally regulate $HAL$ expression (Supplementary Fig. 2e).

To investigate whether expression of the three factors was regulated in an Wnt/β-catenin-dependent manner in liver cancer, we treated HuH-7 cells with a GSK-3α/β inhibitor, CHIR-99021, and examined their expression. As a result, the treatment significantly down-regulated the expression of CEBPA, FOXA1, and FOXA3 (Fig. 1d). In complete agreement with these results, TCGA data showed that HCCs carrying mutant $CTNNB1$ have decreased expression of the three TFs as well as $HAL$ compared to those with the wild-type (Fig. 1e). Taken together, these data strengthened our findings that CEBPA, FOXA1, and FOXA3 are down-regulated by the Wnt signaling.

### The effect of CEBPA, FOXA1, and FOXA3 on the reporter activity of HAL

Next, we examined the interaction of the three TFs with the regulatory region between −90 bp and −44 bp of $HAL$. The JASPAR database suggested two CEBP-binding elements (CBE-1 and CBE-2) and a Forkhead-binding element (FBE) within the region to interact with both FOXA1 and FOXA3 (Supplementary Fig. 3a, b). To analyze whether these three elements are functionally associated with the transcriptional activity, we used a reporter plasmid containing the $HAL$ promoter and its minimal regulatory region upstream of the firefly luciferase gene (pHAL-90/+147)[7]. We generated four mutant reporter plasmids (Fig. 2a), namely a three-base substitution in CBE-1 (CBE-1m), a two-base substitution in CBE-2 (CBE-2m), a three-base substitution in FBE (FBEm), and a combination of the three mutations (CBE-1m, CBE-2m, and FBEm). Reporter assay using the wild-type plasmids in combination with β-catenin siRNA confirmed that knockdown of β-catenin increased the wild-type reporter activity in HepG2 cells (Fig. 2b). The increase in wild type reporter activity by the β-catenin siRNA was partially reduced by the mutations of three binding motifs (CBE-1m, CBE-2m, and FBEm). Notably, the combination of the mutations in the three motifs almost abolished the increased reporter activity by the β-catenin siRNA. We additionally analyzed the reporter activity of the wild-type plasmids by over-expression of the three TFs in HepG2 cells. As a result, over-expression of CEBPA alone significantly augmented the reporter activity (Fig. 2c and Supplementary Fig. 4a). Furthermore, knockdown of CEBPA or FOXA1 with two-independent siRNAs for each gene significantly decreased the reporter activity, but the activity was not changed by the knockdown of FOXA3 (Fig. 2d).

### CEBPA and FOXA1 are bona fide regulators of the expression of HAL

To further investigate the involvement of these three TFs, we treated HuH-7 and Hep3B cells with CEBPA, FOXA1, or FOXA3 siRNAs and

 

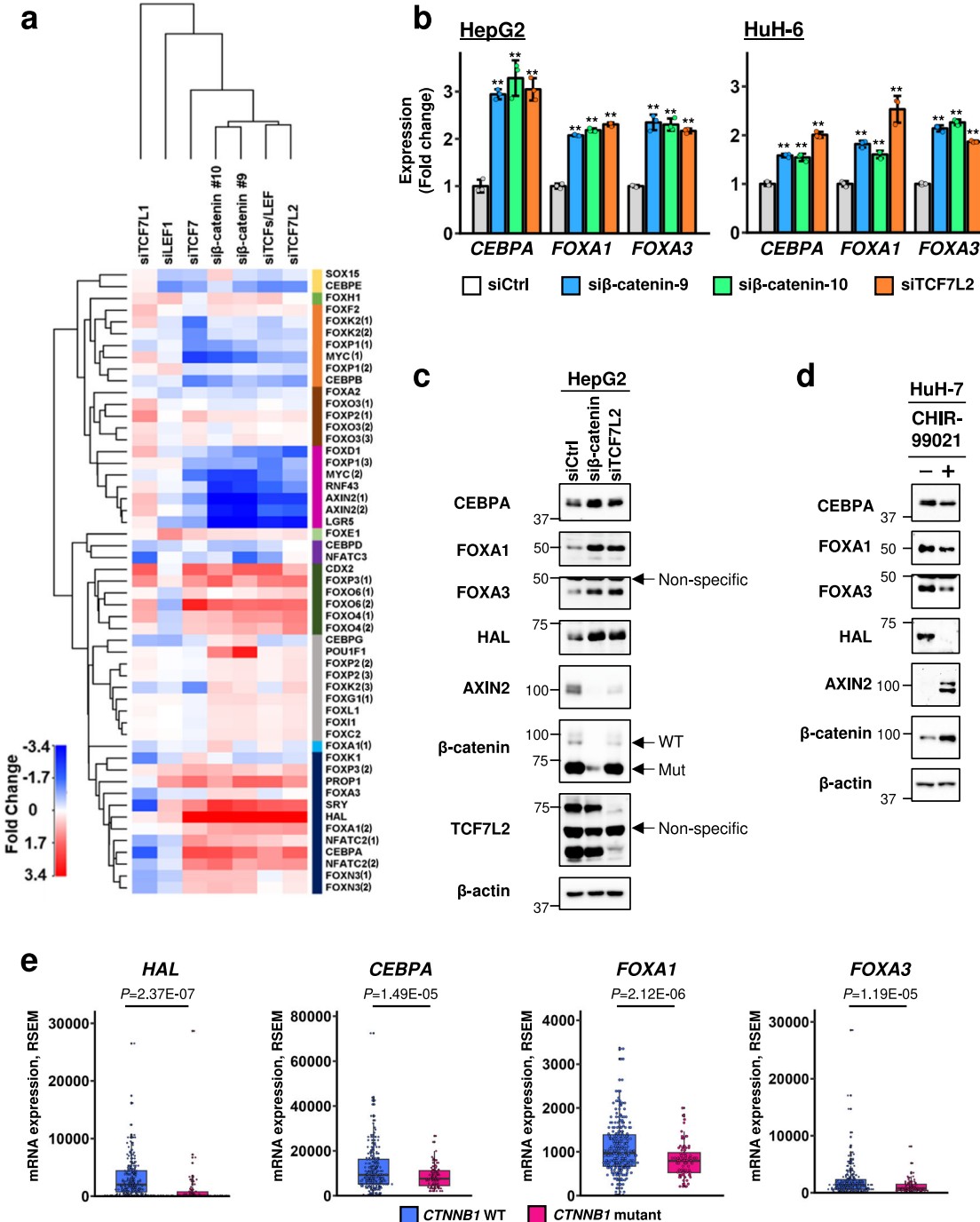

**Fig. 1 | Identification of transcription factors that are negatively regulated by the β-catenin-TCF/LEF complex in liver cancer. a** Hierarchical cluster analysis of HepG2 cells treated with siRNA targeting components of β-catenin-TCF/LEF complex. Expression levels of genes encoding candidate transcription factors, Wnt target genes (*AXIN2*, *LGR5*, *MYC*, and *RNF43*), and *HAL* are shown in the heatmap. The numbers after the gene symbols indicate the different microarray probes. Information of probe ID is shown in Supplementary Data 1c. **b** Expression levels of *CEBPA*, *FOXA1*, and *FOXA3* in HepG2 and HuH-6 cells treated with control, β-catenin (−9 and −10), or TCF7L2 siRNA were analyzed by RT-qPCR. The *y*-axis represents fold change of the expression in the cells treated with β-catenin or TCF7L2 siRNA compared to control siRNA. *HPRT1* was used as an internal control for qPCR. The data represent mean ± SD from three-independent experiments. Statistical significance was determined by Dunnett's test. **p < 0.01 vs siCtrl.

**c** Increased expression of CEBPA, FOXA1, and FOXA3 by β-catenin or TCF7L2 siRNA in HepG2 cells. AXIN2 was used as a direct target of the β-catenin/TCF complex. Since HepG2 cells carry a heterozygous deletion encompassing exon 3 and exon 4 of *CTNNB1*, immunoblot analysis of the cells depicted two bands corresponding to the wild-type (96 kDa) and mutant (75 kDa) β-catenin protein. **d** Effect of CHIR-99021, a GSK3 inhibitor, on the expression of β-catenin, AXIN2, HAL, CEBPA, FOXA1, and FOXA3 in HuH-7 cells. **e** Expression of *HAL*, *CEBPA*, *FOXA1*, and *FOXA3* in hepatocellular carcinoma tissues with or without *CTNNB1* mutation. The expression values and genetic status of *CTNNB1* mutation were obtained from the dataset of 361 hepatocellular carcinoma (TCGA, Pan-Cancer Atlas). Statistical significance was determined by unpaired two-tailed *t*-test. Center line, median; Box limits, upper and lower quartiles; Whiskers, 1.5× interquartile range.

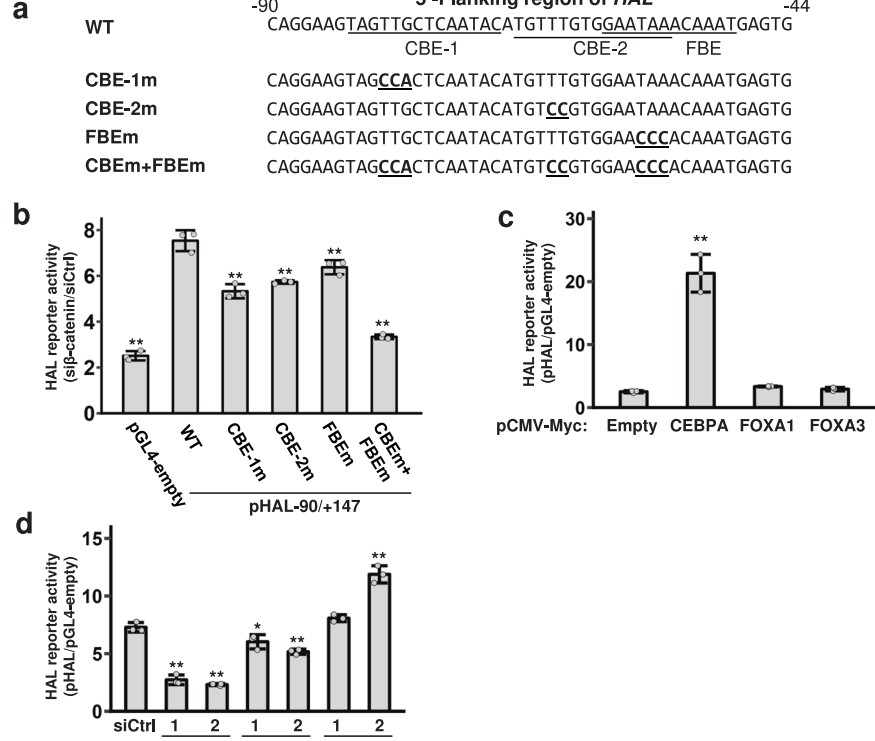

**Fig. 2 | Reporter assay of the *HAL*-promoter region containing putative CEBP- and Forkhead-binding elements. a** Sequences of the wild type and mutant *HAL*-promoter region used for the reporter assay. The 5'-flanking region of *HAL* (between −90 and −44 bp) contains two putative CEBP-binding elements (CBE-1 and CBE-2) and a putative Forkhead-binding element (FBE). Substitutions in the binding elements are underlined and shown in bold. **b** Reporter activity of the wild-type and mutant HAL-reporter plasmids in response to β-catenin-9 siRNA. The activity of each reporter plasmid in the cells treated with β-catenin siRNA was divided by that with control siRNA. Dual reporter assay was carried out using pRL-null plasmid for the normalization of transfection. **c** The effect of CEBPA, FOXA1, and FOXA3 over-expression on the reporter activity. HepG2 cells were transfected with wild type HAL-reporter plasmid and the indicated transcription factors. The *y*-axis represents relative reporter activity compared to the mock reporter plasmid. **d** The effect of siRNA for CEBPA, FOXA1, and FOXA3 on the reporter activity. Hep3B cells transfected with the wild type reporter plasmid were treated with the indicated siRNAs. The *y*-axis represents the relative reporter activity compared to the mock reporter plasmid. Unless specified otherwise, data are represented as the mean ± SD of three independent cultures. Statistical significance was determined by Dunnett's test. *$p < 0.05$, **$p < 0.01$ vs Empty or siCtrl.

examined the expression of HAL by quantitative PCR and immunoblot analysis. Consistent with the reporter assay results, knockdown of CEBPA or FOXA1 reduced HAL expression (Fig. 3a, b and Supplementary Fig. 4b), but that of FOXA3 did not show a consistent decrease in the levels of HAL expression in both cell lines. In addition, HAL expression was induced by the overexpression of CEBPA or FOXA1 but not by that of FOXA3 (Fig. 3c, d and Supplementary Fig. 4c). Taken together, our results suggest that CEBPA and FOXA1 play a vital role in the transcriptional activation of *HAL* through the regulatory region located in the 5'-flanking region.

## Identification of downstream targets of CEBPA and FOXA1 in liver cancer cells

To unveil the roles of CEBPA and FOXA1 that were suppressed by the Wnt/β-catenin signaling pathway, we further searched for additional target genes of these two TFs. RNA-seq analysis of HuH-7 cells treated with CEBPA or FOXA1 siRNAs identified genes which were down-regulated by the siRNAs. This analysis corroborated that expression of *HAL* was regulated by both CEBPA and FOXA1 (Supplementary Data 2a, b). We further performed ChIP-seq analysis and obtained significant peaks of the interaction with CEBPA and FOXA1 across the genome (Supplementary Fig. 5a and Supplementary Data 2c, d). It is of note that CEBPA and FOXA1 peaks were observed in the 5'-flanking region of the *HAL* gene (Fig. 4a). An additional ChIP-qPCR analysis verified the interaction of this region with CEBPA and FOXA1 (Supplementary Fig. 5b). Integration of these data identified a total of 460 and 489 potential direct target genes positively regulated by CEBPA and FOXA1, respectively (FDR *q*-values < 0.05, Fig. 4b and Supplementary

Data 3a, b). To elucidate biological roles of the two TFs, we performed enrichment analysis with the potential target genes using the KEGG pathway database. Consequently, 14 and 15 pathways were significantly related to CEBPA and FOXA1 expression, respectively (FDR *q*-values < 0.01, Fig. 4c and Supplementary Data 3c, d). These included common pathways such as "Arginine and proline metabolism" and "Pathways in cancer". In addition, we searched for genes commonly down-regulated by knockdown of CEBPA and FOXA1, and identified a total of 132 genes including *HAL* (Fig. 4d and Supplementary Data 3e). Subsequent pathway analysis with the 132 genes uncovered significant enrichment of a gene set "Arginine and proline metabolism" (FDR *q*-value < 0.01, Fig. 4e and Supplementary Data 3f), and this gene set included four differentially expressed genes, *AMD1*, *ARG1*, *GLS*, and *GOT1* (Supplementary Fig. 6a). To investigate whether these four genes are down-regulated by the Wnt signaling pathway, we analyzed their expression levels in HepG2 cells treated with control, β-catenin, or TCF7L2 siRNA. Among the four, only the expression of *ARG1* was remarkably increased by the suppression of β-catenin or TCF7L2 siRNA compared with control siRNA (Fig. 4f and Supplementary Fig. 6b). Consistent with our results, TCGA data showed that HCCs carrying mutant *CTNNB1* have decreased expression of *ARG1* compared to those with the wild-type (Supplementary Fig. 6c). In addition, knockdown of CEBPA and FOXA1 decreased ARG1 expression (Supplementary Fig. 6d, e). We also found significant CEBPA and FOXA1 peaks in intron 1 of the *ARG1* gene in the ChIP-seq data (Fig. 4g and Supplementary Fig. 5c).

To confirm whether the Wnt/β-catenin signaling regulates *ARG1* through the suppression of CEBPA and FOXA1, we analyzed their

**Fig. 3 | Involvement of CEBPA and FOXA1 in HAL expression. a** Expression of *HAL* in HuH-7 and Hep3B cells treated with the indicated siRNAs. Expression was assessed by RT-qPCR. The *y*-axis represents the fold change in the expression of *HAL* observed in the cells treated with the indicated siR-NAs compared to control siRNA. **b** Expression of HAL, CEBPA, FOXA1, and FOXA3 in the cells treated with the indicated siRNAs was detected by immunoblotting. β-actin served as a loading control. **c** The effect of CEBPA, FOXA1, and FOXA3 over-expression on the expression of *HAL* in HepG2 and Hep3B cells. Expression was assessed by RT-qPCR. The *y*-axis represents fold change in *HAL* expression observed in the cells over-expressing the indicated transcription factors relative to EGFP (control). **d** Expression of HAL, CEBPA, FOXA1, and FOXA3 in the cells over-expressing the indicated plasmids was detected by immunoblotting. β-actin served as a loading control. Unless specified otherwise, data are represented as the mean ± SD of three independent experiments. Statistical significance was determined by Dunnett's test. $^{**}p < 0.01$ vs siCtrl or EGFP.

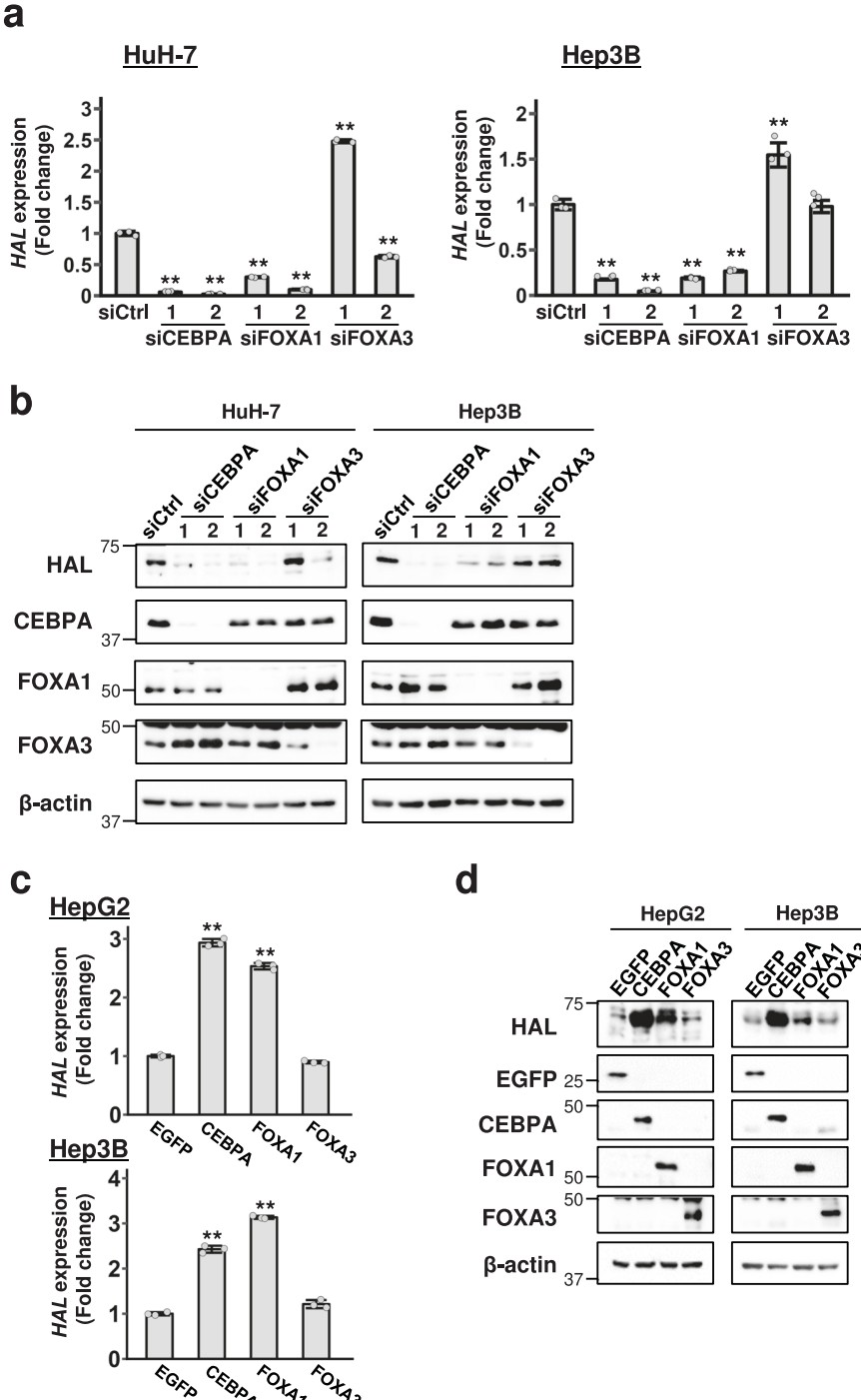

expression levels in HepG2 cells transfected with β-catenin with/without CEBPA or FOXA1 siRNA. As shown in Fig. 4h, knockdown of β-catenin alone increased the expression of CEBPA and FOXA1 as well as that of ARG1. Concomitant suppression of β-catenin and CEBPA or FOXA1 counteracted the induction of ARG1 by β-catenin siRNA. A similar result was obtained for HAL expression. These results suggest that Wnt signaling regulates the expression of HAL and ARG1 through CEBPA and FOXA1.

**Wnt signaling pathway is associated with the levels of cellular amino acids**

The down-regulation of *HAL* and *ARG1* by Wnt signaling is likely dependent on the type of tissue because the expression of *HAL* and *ARG1* did not decrease in the Wnt-activated colorectal adenocarcinoma (Supplementary Fig. 6f). Since these enzymes are involved in the metabolism of histidine or arginine, we examined whether the levels of metabolites are regulated by the Wnt signaling pathway in liver cancer cells. Metabolome analysis was performed using capillary electrophoresis time-of-flight mass spectrometry and the levels of 116 metabolites were compared in the lysates from HepG2 cells transfected with control, β-catenin, or TCF7L2 siRNA. The heatmap depicted that metabolites commonly altered by both β-catenin and TCF7L2 siRNAs were mainly reduced (Fig. 5a and Supplementary Data 4a). Consistent with the increased expression of HAL and ARG1 by the siRNA, the levels of histidine and arginine were significantly decreased in the cells transfected with β-catenin or TCF7L2 siRNA (Fig. 5b). To determine whether the levels of these amino acids were regulated by CEBPA or FOXA1, we analyzed the levels of histidine and arginine in HepG2 cells

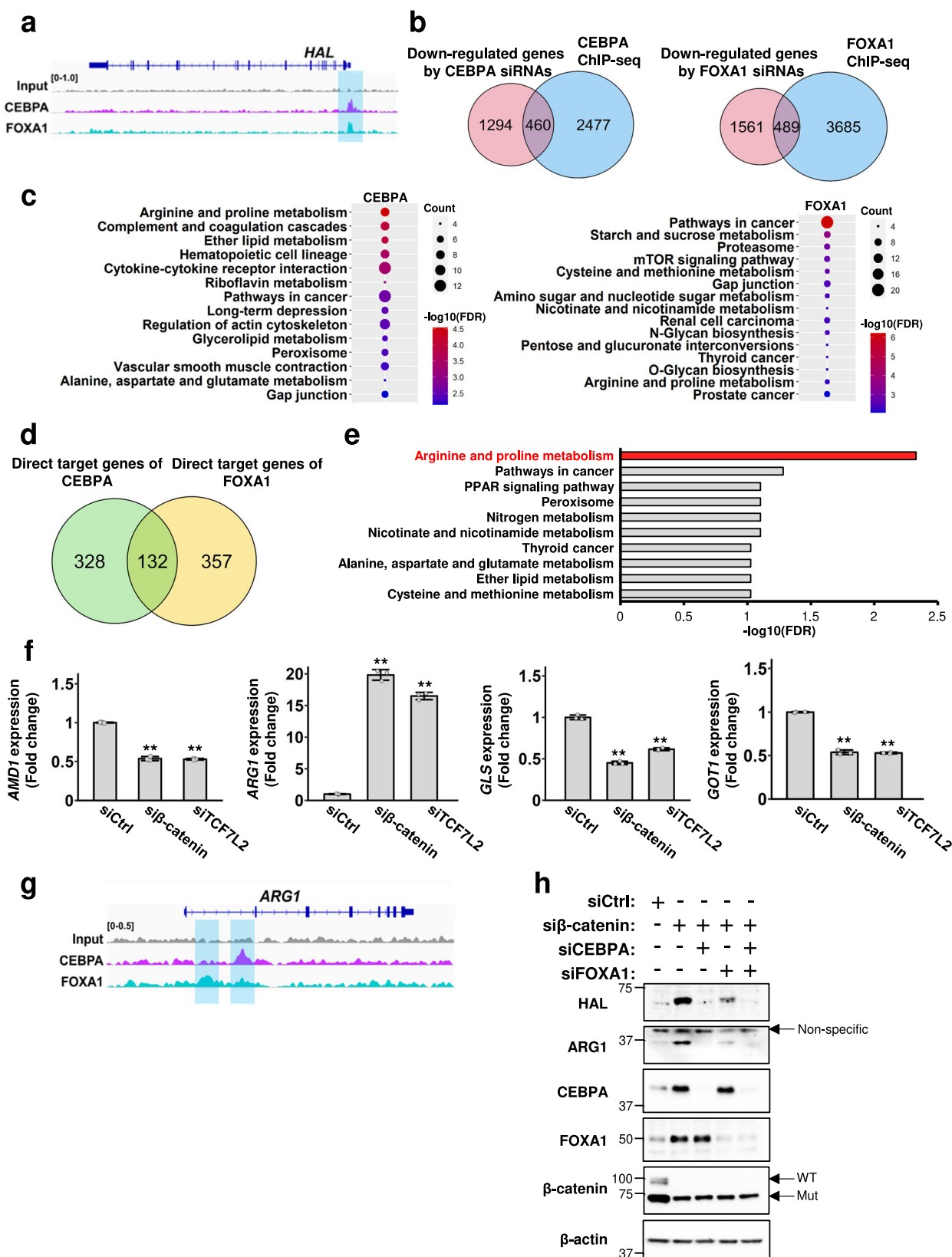

transfected with β-catenin siRNA in combination with CEBPA or FOXA1 siRNA (Supplementary Fig. 7a). As expected, the levels of histidine and arginine were significantly restored by the additional knockdown of CEBPA. There was also a trend toward restoration of histidine and arginine levels in cells treated with β-catenin and FOXA1 siRNAs. These results were

consistent with the western blotting results showing that knockdown of CEBPA and FOXA1 reduced the silencing effect of β-catenin on HAL and ARG1 expression (Fig. 4h). These findings further support the involvement of these transcription factors in cellular amino acid metabolism. Interestingly, inhibition of the Wnt signaling pathway significantly decreased a large

**Fig. 4 | Gene set enrichment analysis using genes directly regulated by CEBPA and FOXA1 in liver cancer cells. a** Visualization of ChIP-seq peaks of CEBPA (magenta) and FOXA1 (blue) in the genomic region of *HAL*. **b** Venn diagrams depicting the number of genes regulated by CEBPA and genes regulated by FOXA1 in HuH-7 cells. The overlapping genes were considered as direct targets of each transcription factor. **c** Over-representation analysis (ORA) using KEGG pathway gene sets with the 460 and 489 genes directly regulated by CEBPA and FOXA1, respectively. Significant pathways are shown with *q*-value and the number of genes. **d** Venn diagram showing common direct targets of CEBPA and FOXA1. **e** ORA

using the 132 common target genes (FDR *q*-value < 0.01). **f** The expression levels of *AMD1*, *ARG1*, *GLS*, and *GOT1* in HepG2 cells treated with β-catenin or TCF7L2 siRNA analyzed by RT-qPCR. *HPRT1* was used as an internal control. Data are represented as the mean ± SD of three independent experiments. Statistical significance was determined by Dunnett's test. **p < 0.01 vs siCtrl. **g** Visualization of ChIP-seq peaks of CEBPA (magenta) and FOXA1 (blue) in the genomic region of *ARG1*. **h** Expression of HAL and ARG1 in HepG2 cells treated with CEBPA and/or FOXA1 siRNA in combination with β-catenin siRNA.

number of amino acids, resulting in the suppression of total levels of amino acids (Supplementary Fig. 7b). In addition, we found that the suppression of the pathway decreased the levels of lactic acid and increased that of acetyl-CoA (Fig. 5c), implying that the inhibition of the Wnt signaling pathway may suppress the Warburg effect in liver cancer cells.

Subsequently, we performed pathway analysis using the data of 41 metabolites that were significantly regulated by β-catenin and TCF7L2 siRNA, and identified the enrichment of 13 metabolite sets (FDR *q*-values < 0.01, Fig. 5d and Supplementary Data 4b). Consistent with the change in the level of arginine, "Arginine and Proline Metabolism" was in the list of the 13 metabolite sets. It is of note that "Urea cycle" was listed as the most significant metabolite set. This result suggests that the activated Wnt signaling pathway may regulate the urea cycle through down-regulation of *ARG1* in hepatoma cells because *ARG1* encodes the focal enzyme of the urea cycle hydrolyzing L-arginine to urea and L-ornithine. We further investigated whether other genes in the urea cycle were modulated by the Wnt signaling pathway using the microarray data (Supplementary Data 1b). In addition to *ARG1*, ornithine transcarbamylase (*OTC*) expression was remarkably increased by the knockdown of β-catenin and TCF7L2 (Fig. 5e), and argininosuccinate synthase 1 (*ASS1*) and argininosuccinate lyase (*ASL*) expression was also substantially increased by the knockdown. These results suggest that the metabolism of amino acids and the urea cycle are altered by the activated Wnt/β-catenin signaling pathway and that these changes may contribute to the development and progression of liver cancer cells.

## Discussion

In this study, we have shown that the promoter region of *HAL* was transcriptionally regulated by transcription factors CEBPA and FOXA1, and that the two were down-regulated by the Wnt signaling pathway in the liver cancer cells. In addition, we have uncovered that the pathway modulates intracellular metabolites at least in part by HAL and ARG1 through suppression of CEBPA and FOXA1.

It was previously reported that *HAL* expression was significantly decreased in the liver of *Hnf4a* (hepatocyte nuclear factor 4α)-knockout mice[20]. In addition, over-expression of β-catenin decreased the expression of *HNF4A* in Hep3B cells[21]. These data led us to additionally investigate whether HNF4A regulates *HAL* in the same manner as CEBPA and FOXA1. The knockdown of β-catenin and TCF7L2 increased the expression of *HNF4A* (Supplementary Fig. 1d). Although knockdown of HNF4A significantly decreased the expression of *HAL* (Supplementary Fig. 1e), it did not affect the reporter activity of the *HAL* promoter (Supplementary Fig. 1f). Therefore, *HAL* expression is regulated by at least three transcription factors, CEBPA, FOXA1, and HNF4A in liver cells, and CEBPA and FOXA1 function through the interaction with the proximal promoter region of *HAL*.

FOXA1 or hepatocyte nuclear factor 3α (HNF3A), a liver-enriched transcription factor, was reported to decrease the transcription of *PIK3R1*, and suppress the viability and motility of HCC cells through the inhibition of PI3K-Akt signaling[22]. In addition, FOXA1 positively regulates miR-122 that is specifically suppressed in liver tumors with poor prognosis[23]. These reports suggest that FOXA1 may function as a tumor suppressor in HCC, and their results are consistent with our finding that the expression of FOXA1 was suppressed in liver cancer cells with activated Wnt signaling. In our reporter gene assay, overexpression or suppression of FOXA1 had limited effect on the activity of the *HAL* promoter (Fig. 2), but the expression

levels of *HAL* were dependent on FOXA1 expression (Fig. 3). This discrepancy may be explained by the fact that FOXA1 is a pioneer factor, which recognizes specific DNA sequences exposed on the surface of a nucleosome and allows other transcription factors and histone modifiers to access silent genes that are inaccessible to general transcription factors[24,25]. Our results indicated that alteration of chromatin structure by FOXA1 might be essential for the regulation of the transcription of *HAL*. Alternatively, a regulatory region other than −90 and −44 bp may be involved in the regulation of *HAL* transcription.

Microarray expression analysis of HepG2 cells with β-catenin or TCF/LEF siRNA confirmed that known Wnt target genes including *MYC*, *RNF43*, *AXIN2*, and *LGR5* were listed as down-regulated genes (Fig. 1a). Several members of FOX-family transcription factors such as *FOXD1* and *FOXP1* were also similarly reduced by these siRNAs, suggesting that *FOXD1* and *FOXP1* may be targets of the canonical Wnt signaling pathway. It is of note that FOXD1 is aberrantly overexpressed in colorectal cancer, and it promotes the progression of cancer through the activation of the ERK1/2 signaling pathway[26]. Regarding FOXP1, it has a paradoxical role in tumorigenesis, depending on the type of cancer[27]. Both FOXD1 and FOXP1 have been reported as activators of Wnt/β-catenin signaling in prostate cancer and diffuse large B cell lymphoma, respectively[28,29]. Therefore, these factors may mediate a positive feedback loop in the Wnt signaling pathway and accelerate the progression of Wnt-driven cancer. The role of these factors in liver carcinogenesis needs to be clarified in future studies.

CEBPA is a transcription factor that is abundantly expressed in the liver, skin, mammary gland, and adipose tissue (GTEx Portal; https://gtexportal.org/home/). The repression of CEBPA was shown in a wide range of liver pathologies including liver fibrosis[30] and cirrhosis[31]. It has been reported that the suppression of CEBPA was caused by the activation of Wnt signaling in mouse embryonic fibroblasts[32] and liver cancer cells[33] at the protein levels. Reportedly, CEBPA protein was degraded by the interaction with tribbles homolog 2 (TRIB2) and constitutive photomorphogenesis 1 (COP1)[34] or tripartite motif-containing protein 21 (TRIM21)[35], and TRIB2 was induced by activation of Wnt/TCF signaling[33]. However, we showed in this study that *CEBPA* was transcriptionally reduced by the activation of Wnt signaling. Because the expression of *FOXA1* and *CEBPA* was not induced by the knockdown of β-catenin in colorectal cancer cells, transcription factors that are expressed in liver tissues might be involved in the regulation of *CEBPA* and/or *FOXA1* expression.

*HAL* and *ARG1*, the two liver-specific Wnt target genes, encode enzymes associated with amino acid metabolisms. HAL catalyzes the first reaction in histidine catabolism[36]. Reportedly, inhibition of the components of the histidine degradation pathway such as HAL and AMDHD1 (amidohydrolase domain containing 1) induces the levels of tetrahydrofolate, which decreases the sensitivity of hematopoietic cancer cells to methotrexate[8]. In addition, the sensitivity to methotrexate in lung cancer cells was decreased by the knockout of *HAL*[8]. These data suggest a link between chemoresistance of the cells and Wnt-activation. Therefore, the Wnt/β-catenin pathway may be involved in chemoresistance and cell differentiation through the suppression of histidine catabolism in hepatoma cells. As for ARG1, which is abundantly expressed in the liver, it cleaves arginine to urea and ornithine in the urea cycle[37]. In this study, we showed that suppressed Wnt signaling changed the levels of metabolites as well as genes associated with the urea cycle (Fig. 5d, e). In agreement with our data, proteomics analysis of liver-specific *Apc* knockout mice demonstrated that

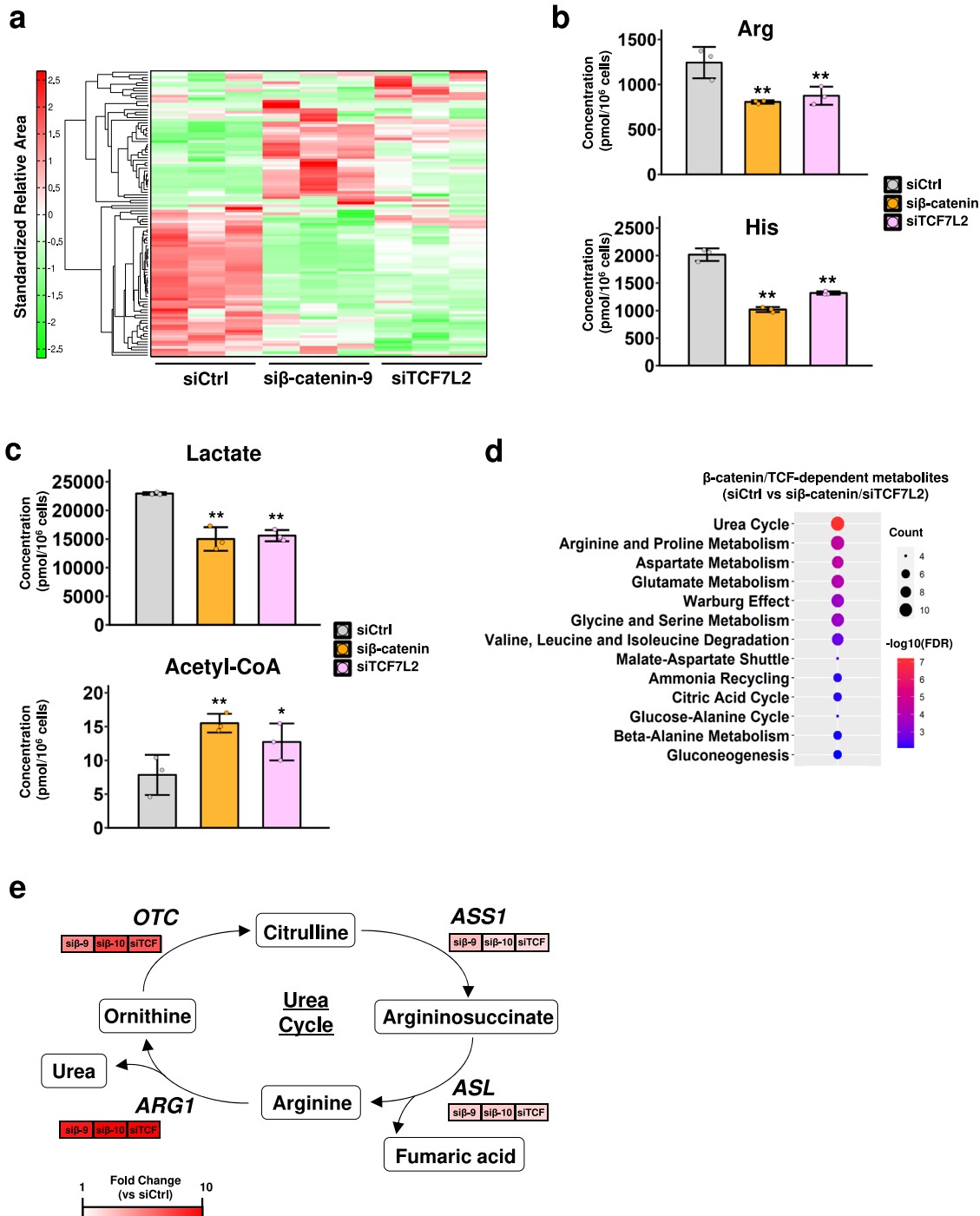

**Fig. 5 | Alterations in the levels of amino acids in β-catenin- or TCF7L2-depleted cells. a** Metabolite analysis using HepG2 cells treated with control, β-catenin, or TCF7L2 siRNA. Each group was analyzed in triplicate. **b, c** Quantitative analysis of the indicated metabolites using mass spectrometry in the HepG2 cells treated with control, β-catenin, or TCF7L2 siRNA. The y-axis represents the concentration (pmol/10⁶ cells) of metabolites. **b** Levels of arginine and histidine. **c** Levels of lactate and acetyl-CoA. **d** Metabolic pathway analysis using metabolites that were commonly altered by the knockdown of β-catenin and TCF7L2. FDR *q*-value < 0.01 was considered significant.

**e** A simplified schematic representation of the urea cycle and related enzymes. *ASS1*, argininosuccinate synthetase1; *ASL*, argininosuccinate lyase; *OTC*, ornithine transcarbamylase. The microarray data (Fig. 1a) were used to generate altered expression values of *OTC*, *ARG1*, *ASS1*, and *ASL* (fold change) in the cells treated with β-catenin (siβ-9 and siβ-10) or TCF7L2 (siTCF) siRNA compared to control siRNA. Unless specified otherwise, data are represented as the mean ± SD of three independent cultures. Statistical significance was determined by Dunnett's test. *\*p* < 0.05, *\*\*p* < 0.01 vs siCtrl.

the most differentially expressed proteins were related to a metabolic pathway, and that the expression of Arg1 and Cps1, urea cycle related enzymes, were suppressed in the knockout mice[38]. Therefore, activated Wnt signaling may accumulate intracellular ammonia by suppression of the urea cycle in liver cancer. The suppression may help the synthesis of various proteins necessary to produce new daughter cells by decreasing the degradation of amino acids[39]. Indeed, our results indicated that suppression of the signaling caused the reduction in the total amount of amino acids (Supplementary Fig. 7b), which may be due to the acceleration of the urea cycle and degradation of amino acids (Fig. 5e). Importantly, it has been reported

that the dysregulation of the urea cycle is widely observed in liver tumors and profoundly affects carcinogenesis, mutagenesis, and immunotherapy response[40]. Targeting the urea cycle might be an attractive therapeutic strategy for patients with liver tumor driven by the activated Wnt pathway.

As shown in Fig. 5c, we observed reduced glycolysis by β-catenin or TCF7L2 siRNAs in HepG2 cells, suggesting that Wnt signaling is associated with glycolysis. It has been reported that Wnt signaling up-regulates pyruvate dehydrogenase kinase-1 (PDK1) in colorectal cancer cells, leading to the enhanced glycolysis through the inhibition of pyruvate dehydrogenase (PDH) activity[11]. Consistent with this report, we confirmed that the expression of *PDK1* was also decreased by the suppression of Wnt signaling in HepG2 cells (Supplementary Data 1b), corroborating that PDK1 plays a pivotal role in glycolysis as a downstream of the Wnt signaling in liver tissue. In cancer cells, increased catabolism of glucose (Warburg effect) anaerobically rather than aerobically leads increased glucose uptake and lactic acid production[41]. It is noteworthy that increased levels of lactic acid by enhanced glycolysis promote immune evasion in tumors[42,43]. Therefore, the activation of the Wnt/β-catenin signaling pathway may, in part, suppress anti-tumor immunity by increased lactic acid production[44].

In conclusion, we identified two transcription factors, CEBPA and FOXA1, enriched in the liver as cellular context-dependent targets of the Wnt signaling. Our findings will provide insights into the relationship between liver metabolism and the Wnt signaling pathway. Further investigations of tissue-specific Wnt targets may uncover a role of the pathway involved in human carcinogenesis.

## Methods
### Cell culture
Human hepatoma cells, HepG2 and Hep3B, were obtained from the American Type Culture Collection (ATCC, Manassas, VA), and HuH-6 and HuH-7 cells were obtained from the Japanese Collection of Research Bioresources Cell Bank (JCRB, Osaka, Japan). These cells were confirmed to be mycoplasma-free and authenticated by Hoechst DNA staining and short tandem repeat profiling, respectively (ATCC and JCRB). Retrovirus packaging cells PLAT-A were provided from Dr. Toshio Kitamura (The University of Tokyo). All cells were grown in appropriate media containing 10% fetal bovine serum (Biosera, Nuaille, France) and 1% penicillin/streptomycin solution (Fujifilm Wako Pure Chemical, Osaka, Japan). The PLAT-A cells were maintained in complete medium supplemented with puromycin (1 µg/ml, Merck, Darmstadt, Germany) and blasticidin (10 µg/ml, Fujifilm Wako Pure Chemical).

### Expression plasmids
The entire coding region of *FOXA1*, *FOXA3*, and *HNF4A* were amplified by RT-PCR (KOD One, Toyobo, Osaka, Japan) using human liver cDNA as template. The coding region of *CEBPA* was amplified using MSCV-CEBPA plasmid (a kind gift from Dr. Atsushi Iwama, The University of Tokyo). The PCR products were cloned into pCMV-Myc-N vector (Takara Bio, Shiga, Japan). The primer sequences used for the amplification are listed in Supplementary Data 5a. The cloned DNA fragments in the plasmids were confirmed by Sanger sequencing (3500xL GeneticAnalyzer, Thermo Fisher Scientific, Waltham, MA).

### Gene silencing
Two independent siRNAs targeting each gene were used for gene silencing. CEBPA, FOXA1, FOXA3, and HNF4A siRNAs were purchased from Integrated DNA Technologies (Coralville, IA). β-catenin, LEF1, TCF7, TCF7L1, and TCF7L2 siRNAs were obtained from Merck. Control siRNA (ON-TARGETplus Non-targeting Pool) was purchased from Horizon Discovery (Cambridge, UK). The target sequences of the siRNAs are shown in Supplementary Data 5b. Cells were transfected with the siRNAs (10 nM) for 48 h using Lipofectamine RNAiMAX (Thermo Fisher Scientific). The gene-silencing efficiency of each siRNA was confirmed by quantitative RT-PCR (qRT-PCR) or immunoblotting.

### Quantitative PCR
Total RNA was isolated from cultured cells using RNeasy Mini Kit (Qiagen, Valencia, CA). cDNA was synthesized from 1 µg of total RNA using ReverTraAce (Toyobo). qPCR was performed using KAPA SYBR Fast qPCR Kit (Kapa Biosystems, Wilmington, MA) and StepOnePlus (Thermo Fisher Scientific) with sets of primers listed in Supplementary Data 5a. The levels of transcripts were determined using the relative standard curve method, and hypoxanthine phosphoribosyl transferase 1 (*HPRT1*) was used as an internal control.

### Immunoblotting
Cells were lysed in radioimmunoprecipitation assay buffer (50 mM Tris-HCl, pH 8.0, 150 mM NaCl, 0.5% sodium deoxycholate, 1% Nonidet P-40, 0.1% sodium dodecyl sulfate) supplemented with a Protease Inhibitor Cocktail Set III (Merck). The proteins were separated by SDS-PAGE and transferred to a nitrocellulose membrane (Cytiva, Marlborough, MA). After blocking with 5% skim milk powder in TBS-T (Tris-buffered saline-Tween20), the membranes were incubated with anti-HAL (Merck, Cat# HPA038547, 1:1000), anti-β-catenin (Cell Signaling Technology, Danvers, MA, Cat# 9582, 1:1000), anti-TCF7L2 (Merck, Cat#05-511, 1:1000), anti-AXIN2 (Cell Signaling Technology, Cat# 2151, 1:1000), anti-CEBPA (Thermo Fisher Scientific, Cat# PA5-77911, 1:1000), anti-FOXA1 (Merck, Cat# 17-10267, 1:1000 and Santa Cruz Biotechnology, Santa Cruz, CA, Cat# sc-101058, 1:1000), anti-FOXA3 (Abcam, Cambridge, UK, Cat# ab108454, 1:1000), anti-ARG1 (Santa Cruz Biotechnology, Cat# sc-365547, 1:1000), anti-GFP (Santa Cruz Biotechnology, Cat# sc-9996, 1:1000), or anti-β-actin (Merck, Cat# A5441, 1:2500) antibody overnight at 4 °C. Horseradish peroxidase-conjugated anti-mouse (NA931V, Cytiva), anti-rabbit IgG (NA9340V, Cytiva), or VeriBlot for IP Detection Reagent (HRP) (ab131366, Abcam) served as the secondary antibody. The membranes were incubated with ImmunoStar LD (Fujifilm Wako Pure Chemical) or SuperSignal West Pico Chemiluminescent Substrate (Thermo Fisher Scientific), and the chemiluminescence images were captured using Amersham Imager 600 system (Cytiva) or ChemiDoc XRS+ System (Bio-rad, Hercules, CA). β-actin was used as a loading control. Probing with a loading control was performed in parallel with the target antibody by cutting the membrane prior to antibody incubation.

### Site-directed mutagenesis
Putative FOX and/or CEBP transcription factor binding motifs in the 5'-flanking region of the *HAL* gene (pHAL −90/+147) were mutated by site-directed mutagenesis[7]. The 5'-flanking region of *HAL*, between −90 bp and +147 bp, was amplified using KOD-Plus-Neo (Toyobo) with each primer set (Supplementary Data 5a). The PCR products were digested with *Dpn* I (Takara Bio) for 2 h at 37 °C, followed by transformation into *E. coli*. Successful mutagenesis was confirmed by Sanger sequencing.

### Treatment with GSK-3α/β inhibitor
A GSK-3α/β inhibitor, CHIR-99021, was purchased from MedChemExpress (Monmouth Junction, NJ), and dissolved in dimethyl sulfoxide (DMSO). HuH-7 cells were treated with CHIR-99021 (5 µM) or DMSO for 24 h.

### Luciferase reporter assays
HepG2 and Hep3B cells seeded on 12-well plates were transfected with 0.25 µg of reporter plasmids and 0.05 µg of pRL-null (Promega, Madison, WI) in combination with the indicated siRNAs using Lipofectamine 2000 reagent (Thermo Fisher Scientific). After 48 h, the cells were harvested, and reporter activity was measured by dual luciferase assay system (TOYO B-Net, Tokyo, Japan). To examine the effect of FOXA1, FOXA3, and CEBPA on the activity of *HAL* promoter, HepG2 cells were co-transfected with HAL reporter plasmid (0.2 µg) and pRL-null plasmid (0.05 µg) in combination with plasmids expressing FOXA1 (0.2 µg), FOXA3 (0.2 µg), or CEBPA (0.02 µg) using FuGENE6 reagent (Promega). Firefly luciferase activity was normalized to *Renilla* luciferase activity (pRL-null).

### Retroviral transduction

*CEBPA*, *FOXA1*, and *FOXA3* were sub-cloned into pMXs-Puro retroviral vector (a kind gift from Dr Kitamura, The University of Tokyo) by the Gibson Assembly cloning method (NEBuilder HiFi DNA Assembly Master Mix, New England Biolabs, Ipswich, MA). The sequences of primers used for the amplification are shown in Supplementary Data 5a. For production of retroviral particles, PLAT-A packaging cells were transfected with pMXs-EGFP (enhanced green fluorescent protein), pMXs-CEBPA, pMXs-FOXA1, or pMXs-FOXA3 for 24 h. After changing the medium, the cells were further incubated for 24 h, and supernatants containing the retrovirus were collected and used for infection. Forty-eight hours after infection, HepG2 cells were selected in the medium containing 1.625 µg/ml of puromycin, and Hep3B and HuH-7 cells were selected with 2.5 µg/ml of puromycin.

### Microarray analysis

Total RNA was isolated from HepG2 cells treated with β-catenin, TCF7, TCF7L1, TCF7L2, or LEF1 siRNA (10 nM) for 48 h using Lipofectamine RNAiMAX. Total RNA was extracted using the RNeasy Plus Mini kit (Qiagen), and subsequently expression profiles were analyzed by SurePrint G3 Gene Expression 8 × 60K microarray (Agilent Technologies, Santa Clara, CA) according to the manufacturer's protocol. Data processing and unsupervised hierarchical clustering were performed using GeneSpring GX14.1 software (Agilent Technologies). In hierarchical clustering, Pearson's center and centroid linkage were used as distance metric and linkage function, respectively.

### RNA-seq analysis

HuH-7 cells were transfected with control, FOXA1, or CEBPA siRNAs (10 nM) for 48 h. Total RNA was extracted from the cells using the RNeasy Plus Mini kit. Agilent Bioanalyzer device (Agilent Technologies) was used to assess the quality of extracted RNA. Subsequently, RNA-seq libraries were prepared with total RNA (100 ng) using an NEBNext Ultra II Directional RNA Library Prep Kit (New England Biolabs) according to the manufacturer's protocol. The libraries were sequenced with 100 bp paired-end reads on the DNBSEQ-G400RS (MGI Tech, Shenzhen, China). The analysis of sequencing data was performed by a standard RNA-seq analytical pipeline. Briefly, STAR(v2.7.3a)[45] was used to align the sequencing data to the human genome (hg38). Quantification of gene expression was performed using RSEM (v1.3.3)[46]. The DESeq2 package (v1.26.0)[47] was used to normalize the read count data and test for differential gene expression.

### Chromatin immunoprecipitation followed by high-throughput sequencing (ChIP-seq)

ChIP-seq was performed using anti-CEBPA (Thermo Fisher Scientific, Cat# PA5-77911) or anti-FOXA1 antibody (Merck, Cat# 17-10267)[48]. HuH-7 cells were cross-linked with 1% formaldehyde for 10 min at room temperature, and 0.1 M glycine was added to quench the formaldehyde. Chromatin extracts were sheared by micrococcal nuclease digestion (New England Biolabs), and protein-DNA complexes were immunoprecipitated with 5 µg of anti-CEBPA or anti-FOXA1 antibody bound to Dynabeads Protein G (Thermo Fisher Scientific). Normal rabbit IgG (Santa Cruz Biotechnology) was used as a negative control. De-crosslinking was performed at 65 °C overnight, and samples were subsequently treated with RNase A (Merck) for 2 h at 37 °C and Proteinase K (Merck) for 30 min at 55 °C. The precipitated protein-DNA complexes were purified by the conventional DNA extraction method, and the purified DNA was used for preparation of sequence libraries. Concentration of input and ChIP'd DNA was measured using Qubit dsDNA HS Assay kit (Thermo Fisher Scientific). ChIP-seq libraries were prepared with 1 ng of DNA using an NEBNext Ultra II DNA Library Prep kit (New England Biolabs) according to the manufacturer's protocol. The libraries were sequenced with 150 bp paired-end reads on the DNBSEQ-G400RS. The sequencing data were aligned to human genome (GRCh38) using Bowtie2 (v2.4.1)[49]. Peak calling followed by assignation

of peaks to genes was performed using MACS2 (v3.6)[50] and HOMER (v4.11)[51]. Peaks with *q*-value < 0.05 were considered significant.

To validate the results of ChIP-seq analysis, ChIP followed by quantitative PCR (ChIP-qPCR) was performed using KAPA SYBR Fast qPCR Kit (Kapa Biosystems) and StepOnePlus (Thermo Fisher Scientific)[52]. The precipitated DNAs were subjected to qPCR analysis using sets of primers encompassing genomic regions with peaks. As negative control non-immune IgG and control primers to amplify exon 1 of the glyceraldehyde-3-phosphate dehydrogenase (*GAPDH*) were used. The primer sequences used are listed in Supplementary Data 5a.

### Metabolite extraction and analysis

After aspiration of culture medium, the cells were washed twice with 5% mannitol solution, and incubated with methanol at room temperature for 30 s to suppress enzyme activity. Next, internal standards (H3304-1002, Human Metabolome Technologies, Yamagata, Japan) were added to the cell extract, followed by further incubation at room temperature for 30 s. The cell extract was then centrifuged at $2300 \times g$, 4 °C for 5 min, after which the supernatant was centrifugally filtered through a Millipore 5-kDa cutoff filter (UltrafreeMC-PLHCC, Human Metabolome Technologies) at $9100 \times g$, 4 °C for 5 h to remove macromolecules. Subsequently, the filtrate was evaporated to dryness under vacuum and reconstituted in Milli-Q water for metabolome analysis at Human Metabolome Technologies.

Metabolome analysis was conducted according to the C-SCOPE package (Human Metabolome Technologies), using capillary electrophoresis time-of-flight mass spectrometry (CE-TOFMS) for cation analysis and CE-tandem mass spectrometry (CE-MS/MS) for anion analysis based on the methods described previously [53,54]. Peaks were extracted using Master-Hands, automatic integration software (Keio University, Yamagata, Japan)[55] and MassHunter Quantitative Analysis B.04.00 (Agilent Technologies) in order to obtain peak information including m/z, peak area, and migration time (MT). Signal peaks were annotated according to the metabolite database of Human Metabolome Technologies, based on their m/z values and MTs. The peak area of each metabolite was normalized to internal standards, and metabolite concentration was evaluated by standard curves with three-point calibrations using each standard compound. Hierarchical cluster analysis[56] was performed by Human Metabolome Technologies's proprietary MATLAB and R programs, respectively. Detected metabolites were plotted on metabolic pathway maps using VANTED software[57].

### Over-representation analysis (ORA)

The biological significance of the expression and metabolome data was assessed by over-representation analysis. Differentially expressed genes by either CEBPA or FOXA1 siRNAs were subjected to KEGG pathway analysis. Gene sets with FDR *q*-value < 0.01 were considered significant. MetaboAnalyst[58] was used for the analysis of differential levels of metabolites by β-catenin and TCF7L2 siRNA. Metabolite sets with FDR *q*-value < 0.01 were considered significant.

### Statistics and reproducibility

The unpaired two-tailed t-test was used when two independent groups were compared. For groups larger than two, statistical analysis was performed using one-way analysis of variance (ANOVA) with Dunnett's post hoc test. These statistical analyses were performed using the Bell-Curve for Excel software (Social Survey Research Information, Tokyo, Japan). A *p*-value < 0.05 was considered statistically significant. Sample sizes are indicated in the figure legends. Data are displayed with error bars showing mean ± SD and individual samples in a bar graph. In RNA-seq and ChIP-seq analyses, significance level was set at a Benjamini–Hochberg FDR-adjusted *p*-value (i.e., *q*-value) of less than 0.05.

### Reporting summary

Further information on research design is available in the Nature Portfolio Reporting Summary linked to this article.

## Data availability

Microarray (GSE244527), RNA-seq (GSE244526), and ChIP-seq (GSE244525) data generated in this study were deposited in the Gene Expression Omnibus (GEO) database. Plasmids generated in this study were deposited into Addgene (pCMV-Myc-CEBPA: 219393, pCMV-Myc-FOXA1: 219394, pCMV-Myc-FOXA3: 219395, pCMV-Myc-HNF4A: 21939). All data supporting the findings of this study are available within the paper, its Supplementary Figs., and Supplementary Data. The source data for the graphs in this study are provided in Supplementary Data 5c–g. All uncropped blots are provided in Supplementary Figs. 8–16.

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

## Acknowledgements
We thank Yumiko Isobe, Seira Hatakeyama, and Yuqing Huang (The University of Tokyo) for their technical assistance. The super-computing resource was provided by Human Genome Center, The Institute of Medical Science, The University of Tokyo (http://sc.hgc.jp/shirokane.html). This work was supported by JSPS KAKENHI grant number JP20K07563 to K.Y. and Takeda Science Foundation to K.Y.

## Author contributions
Conceptualization: K.Y., S.N., Y.F. Formal Analysis, S.N., K.Y. Investigation: S.N., K.Y. Methodology: S.N., K.Y., K.T., S.T., T.I., Y.F. Visualization: S.N., K.Y. Writing—Original Draft: S.N., K.Y., Y.F. Writing—Review & Editing: S.N., K.Y., K.T., S.T., T.I., Y.F. Funding Acquisition: K.Y. Supervision: K.Y., Y.F.

## Competing interests
The authors declare no competing interests.
