## [Peer Review File · Communications Biology]

Reviewers' comments:

Reviewer #1 (Remarks to the Author):

In the manuscript submitted by Nakagawa S et al., the authors highlight the pivotal role of CEBPA and FOXA1 transcription factors, under the regulation of the Wnt/ β -catenin pathway, as central regulators of histidine ammonia lyase (HAL) expression. The elucidation of these regulatory mechanisms offers valuable insights into the modulation of amino acid metabolism by β -catenin and its significance in hepatocellular carcinoma (HCC) progression. Additionally, the authors identify Arginase 1 (ARG1) as another target regulated by these molecules, providing a comprehensive understanding of cellular metabolism regulation mediated by Wnt/ β -catenin signaling. The authors employed transcriptome and reporter assays, complemented by ChIP-seq analyses, to demonstrate the direct association of these factors with DNA and their modulation of target gene expression. The study's conceptualization holds significant value, and the experiments are well-designed, offering detailed insights into the role of transcription factors. Metabolite analyses further solidify the association between HAL and ARG1 expression and cellular metabolism.

This reviewer the publication of the manuscript has merit, pending the following comments are addressed:

- 1- While the data for FOXA1 is convincing, there is ambiguity regarding the partial effect of siRNA on expression compared to the lack of effect observed with overexpression on reporter activity. Further explanation into other regulatory regions may help elucidate this discrepancy.
- 2- ChIP-qPCR validation studies where appropriate would enhance the robustness of the findings.
- 3- Consideration of incorporating functional rescue experiments to provide additional strength to the conclusions drawn from the study is essential.

Reviewer #2 (Remarks to the Author):

Saya Nakagawa et al report the identification of transcription factors CEBPA and FOXA1 were responsible for the regulation to HAL in Wnt/ β -catenin signaling. They also revealed these two factors also increased the expression of arginase 1 (ARG1) that catalyzes the hydrolysis of arginine. Metabolome analysis disclosed that activated Wnt signaling augmented intracellular concentrations of histidine and arginine, and that the signal also increased the level of lactic acid suggesting the induction of the Warburg effect in liver cancer cells. Overall, the data is convincing and interesting. With modification it I believe it will make a valuable contribution to the HAL and Wnt signaling in metabolism.

Specific queries/comments that should be addressed:

1. The clustering analysis in fig.1A was a little confused to me. FOXK2 was repeated twice, so did FOXO3, FOXO4, FOXP2, MYC...
2. Line 31-32 in Page5, the author suggested that "These data suggested that CEBPA, FOXA1, and FOXA3 were candidates that transcriptionally regulate HAL expression." Actually, data from fig.1A-C just implied the regulation of β -catenin on CEBPA, FOXA1, and FOXA3.
3. The lower molecular weight band (~75KD) in Fig.1C was indicated by authors as mutated β -catenin. Why? There was no study yet reporting the splicing of CTNNB1.
4. The HAL reporter should be briefly introduced in the second part of Results.
5. The format should be consistent, especially italic or not, such as MYC and PDK1 in Line36 of Page3; HAL in Line 7-11 of Page3, " β " in Line 26-27 in Page5...
6. In general the article would benefit from language editing prior to publication to improve readability.

COMMENT TO THE EDITOR AND THE REVIEWERS

Thank you for giving us the opportunity to submit a revised draft of our manuscript. We are grateful to the reviewers for their careful reading of the manuscript and their constructive comments, which helped us to improve our manuscript. We have revised the issues brought up by the reviewers and added new data. Please find the point-by-point response and the changes made to the text described below.

ANSWERS TO THE REVIEWERS

Reviewer #1:

In the manuscript submitted by Nakagawa S et al., the authors highlight the pivotal role of CEBPA and FOXA1 transcription factors, under the regulation of the Wnt/ β -catenin pathway, as central regulators of histidine ammonia lyase (HAL) expression. The elucidation of these regulatory mechanisms offers valuable insights into the modulation of amino acid metabolism by β -catenin and its significance in hepatocellular carcinoma (HCC) progression. Additionally, the authors identify Arginase 1 (ARG1) as another target regulated by these molecules, providing a comprehensive understanding of cellular metabolism regulation mediated by Wnt/ β -catenin signaling. The authors employed transcriptome and reporter assays, complemented by ChIP-seq analyses, to demonstrate the direct association of these factors with DNA and their modulation of target gene expression. The study's conceptualization holds significant value, and the experiments are well-designed, offering detailed insights into the role of transcription factors. Metabolite analyses further solidify the association between HAL and ARG1 expression and cellular metabolism.

Reply to Reviewer #1:

We are delighted with your high opinion of our manuscript. Thank you for your comments, which have helped us to improve the manuscript.

Comment 1:

While the data for FOXA1 is convincing, there is ambiguity regarding the partial effect of siRNA on expression compared to the lack of effect observed with overexpression on reporter activity. Further explanation into other regulatory regions may help elucidate this discrepancy.

Response to comment 1:

As the reviewer noticed, FOXA1 clearly affected the expression of HAL (Fig.3A-C), whereas it had a marginal effect on the activity of the *HAL* reporter (Fig.2C, D). FOXA1 has been known to be a pioneer factor that initiates cooperative interaction with other regulatory proteins to induce changes in chromatin structure. Thus, we assumed that FOXA1 could not significantly alter the promoter activity using the reporter plasmids. In addition, additional regulatory region(s) may exist outside of -90 and -44 bp in the *HAL* gene. These possibilities are mentioned in the "Discussion" section of the revised manuscript (line 9-18, page 10).

Comment 2:

ChIP-qPCR validation studies where appropriate would enhance the robustness of the findings.

Response to comment 2:

To validate the ChIP-seq results, we performed ChIP-qPCR experiments. DNA fragments precipitated with anti-CEBPA or anti-FOXA1 antibody were subjected to qPCR analysis using primer sets for the promoter region of the *HAL* gene or intron 1 of the *ARG1* gene. A primer set for exon 1 of the *GAPDH* gene was used as negative control.

DNA fragments containing the CEBPA- and FOXA1-binding regions identified by the ChIP-seq analysis were significantly enriched in the precipitants. We have included these data in the “Materials and methods” (line 7-12, page 16) and the “Results” sections (line 11-12, page 7), and in Fig.S5B in the revised manuscript. The primer sequences used have been added in Table S10.

Comment 3:

Consideration of incorporating functional rescue experiments to provide additional strength to the conclusions drawn from the study is essential.

Response to comment 3:

To examine whether knockdown of CEBPA and FOXA1 rescues the decrease of histidine and arginine by the silencing effect of β -catenin on the levels, we performed additional metabolome analyses using HepG2 cells treated with β -catenin siRNA in combination with CEBPA or FOXA1 siRNA. We confirmed that the levels of histidine and arginine were decreased by the knockdown of β -catenin. As expected, the levels of histidine and arginine were significantly restored by the additional knockdown of CEBPA. There was also a trend toward restoration of histidine and arginine levels in cells treated with β -catenin and FOXA1 siRNAs. These results were consistent with the western blotting results showing that knockdown of CEBPA and FOXA1 reduced the silencing effect of β -catenin on *HAL* and *ARG1* expression (Fig. 4H). Since these data further support the involvement of these transcription factors in the cellular metabolism of amino acids, we have included the data in the “Results” section (line 19-28, page 8) and as a new Fig. S7A.

Reviewer #2:

Saya Nakagawa et al report the identification of transcription factors CEBPA and FOXA1 were responsible for the regulation to *HAL* in Wnt/ β -catenin signaling. They also revealed these two factors also increased the expression of arginase 1 (*ARG1*) that catalyzes the hydrolysis of arginine. Metabolome analysis disclosed that activated Wnt signaling augmented intracellular concentrations of histidine and arginine, and that the signal also increased the level of lactic acid suggesting the induction of the Warburg effect in liver cancer cells. Overall, the data is convincing and interesting. With modification it I believe it will make a valuable contribution to the *HAL* and Wnt signaling in metabolism.

Reply to Reviewer #2:

Thank you for your positive assessment of our manuscript. We appreciate your constructive comments, which have helped us to improve the manuscript.

Comment 1:

The clustering analysis in fig.1A was a little confused to me. FOXK2 was repeated twice, so did FOXO3, FOXO4, FOXP2, MYC...

Response to comment 1:

I apologize for any confusion caused by the lack of explanation. As the microarray contains one or more probes for each gene, duplicate genes appeared in the list (Fig.1A). To resolve the confusion, we have added numbers after the gene symbol (Fig. 1A), and the information of probe IDs corresponding each number is shown in Table 2. These are also described in the revised figure legend (line2-3, page 24).

Comment 2:

Line 31-32 in Page5, the author suggested that “These data suggested that CEBPA, FOXA1, and FOXA3 were candidates that transcriptionally regulate HAL expression.” Actually, data from fig.1A-C just implied the regulation of β -catenin on CEBPA, FOXA1, and FOXA3.

Response to comment 2:

We showed the regulation of CEBPA, FOXA1, and FOXA3 by β -catenin and TCF7L2 in Fig.1A-D. We also demonstrated high correlation of their expression with *HAL* expression in hepatocellular carcinoma (Fig. S2A), and showed abundant expression of CEBPA, FOXA1, and FOXA3 and *HAL* in the liver according to the GTEx Portal database (Fig. S2C and S2D). We believe that these data are enough to focus on the three as candidate transcription factors responsible for the regulation of *HAL* expression. The strategy of selection is summarized in Fig. S2E.

Comment 3:

The lower molecular weight band (~75KD) in Fig.1C was indicated by authors as mutated β -catenin. Why? There was no study yet reporting the splicing of CTNNB1.

Response to comment 3:

I apologize for the insufficient explanation about the low molecular weight protein of β -catenin. Because HepG2 cells carry a heterozygous deletion encompassing exon 3 and exon 4 of the *CTNNB1* gene, immunoblot analysis of the cells depicted two bands corresponding to wild-type (96 kDa) and mutant (75 kDa) β -catenin protein as reported by Yamaguchi et al. (Ref#1) and Zeng G et al. (Ref#2). Since the mutant protein lacks phosphorylation sites (Ser33, Ser37, Thr41, and Ser45) by GSK-3 and casein kinase 1, critical residues for subsequent ubiquitination and degradation, the mutant protein was more stable than the wild type β -catenin protein as shown in Fig. 1C. We have briefly included this information in the revised figure legend (line11-13, page 24).

Reference#1:

Yamaguchi K, Zhu C, Ohsugi T, Yamaguchi Y, Ikenoue T, Furukawa Y. Bidirectional reporter assay using HAL promoter and TOPFLASH improves specificity in high-throughput screening of Wnt inhibitors. *Biotechnol Bioeng*. 2017 Dec;114(12):2868-2882.

Reference#2:

Zeng G, Apte U, Cieply B, Singh S, Monga SP. siRNA-mediated beta-catenin knockdown in human hepatoma cells results in decreased growth and survival. *Neoplasia*. 2007 Nov;9(11):951-9.

Comment 4:

The HAL reporter should be briefly introduced in the second part of Results.

Response to comment 4:

Thank you for your comment. We have added the information of reporter plasmids in the “Results” section (line 11-16, page 6) as follows: “To analyze whether these three elements are functionally associated with the transcriptional activity, we used a reporter plasmid containing the *HAL* promoter and its minimal regulatory region upstream of the firefly luciferase gene (pHAL-90/+147) (Ref#1). We generated four mutant reporter plasmids (Fig. 2A), namely a three-base substitution in CBE-1 (CBE-1m), a two-base substitution in CBE-2 (CBE-2m), a three-base substitution in FBE (FBEm), and a combination of the three mutations (CBE-1m, CBE-2m, and FBEm)”.

Reference#1:

Yamaguchi K, Zhu C, Ohsugi T, Yamaguchi Y, Ikenoue T, Furukawa Y. Bidirectional reporter assay using HAL promoter and TOPFLASH improves specificity in high-throughput screening of Wnt inhibitors. *Biotechnol Bioeng*. 2017 Dec;114(12):2868-2882.

Comment 5:

The format should be consistent, especially italic or not, such as MYC and PDK1 in Line36 of Page3; HAL in Line 7-11 of Page3, “ β ” in Line 26-27 in Page5...

Response to comment 5:

We thank the reviewer’s careful reading. We have corrected the format for gene, transcript and protein name throughout the manuscript.

Comment 6:

In general the article would benefit from language editing prior to publication to improve readability.

Response to comment 6:

In response to the comment, this revised manuscript has been edited by a native speaker.

REVIEWERS' COMMENTS:

Reviewer #1 (Remarks to the Author):

Having thoroughly reviewed the revised version, the authors have addressed my concerns. In essence, the revisions have significantly improved the overall quality of the manuscript. I believe that the manuscript will make a valuable contribution to the journal and to the field at large. Therefore, I am happy to endorse its acceptance for publication.

Reviewer #2 (Remarks to the Author):

Dear Editor,

The manuscript has been carefully revised by the authors, and addressed my concerns in the rebuttal letter. The work is interesting and convincing, and can be accepted.